# Growing Gold Nanostars on SiO$_2$ Nanoparticles: Easily Accessible, NIR Active Core–Shell Nanostructures from PVP/DMF Reduction

**Laura C. Straub** [1] , **John A. Capobianco** [2] **and Mathias S. Wickleder** [1,*]

[1]   Department of Chemistry, Institute of Inorganic Chemistry, University of Cologne, 50939 Cologne, Germany; laura.straub@uni-koeln.de
[2]   Department of Chemistry and Biochemistry, Centre for NanoScience Research, Concordia University, Montreal QC H4B 1R6, Canada; john.capobianco@concordia.ca
*   Correspondence: mathias.wickleder@uni-koeln.de

**Abstract:** A new synthesis strategy towards gold-coated silica nanoparticles is presented. The method provides an efficient, reliable and facile-coating process of well-defined star-shaped shell structures, characterized by UV-Vis, TEM, PXRD, DLS and zeta-potential measurements. A marked red shift of the Au-based plasmonic band to the region of the first biological window is observed offering great potential for future research of biological applications.

**Keywords:** core–shell nanostructures; gold nanostars; spiky shells

## 1. Introduction

Core–shell nanostructures attracted increasing attention in the late 1980s and early 1990s with the principle objectives to enhance the stability of nanoparticles, to functionalize the particles for disease targeting and employ their optical properties, for example in bioimaging and drug delivery [1–4]. A flurry of research activities in the following decades led to numerous nanoconstructs with potential applications in nanobiomedicine [5–9].

In the field of nanocomposites with dielectric cores covered by a metallic shell, gold-coated silica nanoparticles (SiO$_2$@Au) have garnered considerable attention. In 1998, Oldenburg et al., reported a colloid reduction process to synthesize gold nanoshells with different core and shell diameters with extinction wavelengths in the near-infrared (NIR) region. The synthesis is based on a stepwise growth of gold nanoparticles (AuNP) attached to the surface of silica nanoparticles (SiO$_2$-NP) [10]. Subsequently, substantial research has been carried out to obtain easily accessible gold coatings [11–14]. Presently, the scientific community continues work on overcoming the challenges of the long aging time required for the shell formation, to ameliorate the choice of reducing agents and to realize more efficient synthetic pathways.

Anisotropic gold nanostructures and SiO$_2$@Au exhibit similar extinction properties in the NIR [15]. For example, gold nanostars (AuNS) show unique surface plasmonic resonances (SPR) compared to spherical gold NP of the same size, caused by the formation of sharp edges. The red-shifted SPR band allows interaction with NIR light, which has higher tissue penetration as well as a good overlap with the first biological window within the region 600 to 1000 nm. The absorption of tissue components, such as melanin, oxy- and deoxyhemoglobin, reach a local minimum in this region which makes it attractive for biological applications [16]. From this perspective, both AuNS and SiO$_2$@Au are of particular interest for multiple biological applications and surface-enhanced spectroscopy [17–22]. Several synthetic routes for AuNS are known using spherical AuNP seeds [22–24]. The Liz-Marzán group synthesized high-yield AuNS using the reduction properties of *N,N*-dimethylformamide (DMF) combined with poly(vinylpyrrolidone) (PVP)

to prepare PVP-stabilized AuNS as well as AuNS containing a magnetite core [25,26]. In addition to PVP and DMF, other capping and reducing agents were used to synthesize spiky shells around polymer, magnetite as well as SiO₂-coated magnetite nanoparticles [27–30]. Combining the advantages of nanoshells and star-shaped structures, our research led to the investigation of star-shaped gold-coated silica nanoparticles (SiO₂@AuNS) shown in Figure 1.

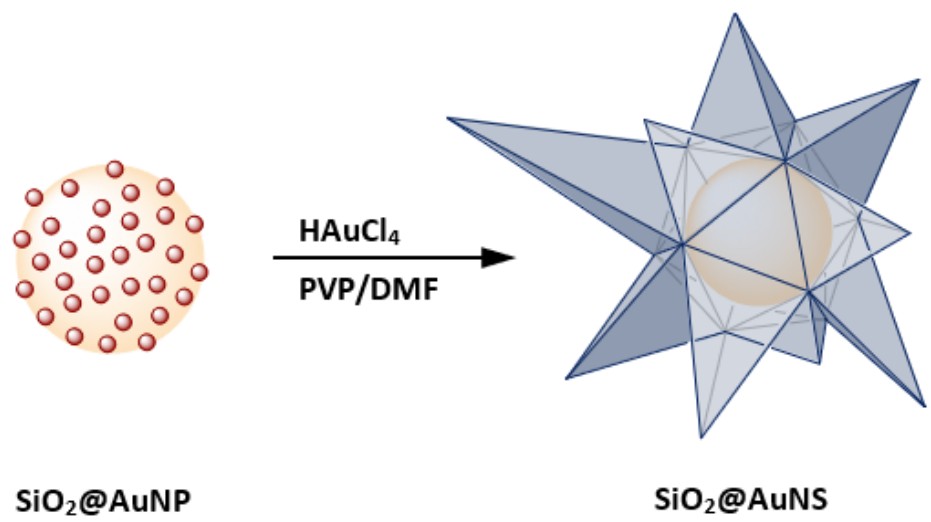

**Figure 1.** A schematic illustration of the SiO₂@AuNS core–shell formation based on the coating of SiO₂@AuNP in PVP/DMF in the presence of HAuCl₄.

Herein, we report the synthesis of a well-defined and size controllable nanostructure, SiO₂@AuNS, using SiO₂-NP cores. The developed synthetic procedure shows a major advantage over previously reported so-called spiky nanoshells because an easy coating approach for various silica-coated nanomaterials becomes accessible [27–29].

## 2. Materials and Methods

All chemicals were used as received without further purification. All glassware were cleaned with aqua regia and rinsed several times with water prior to use (Caution: strong oxidizer). UV-Vis data were recorded with a UV-1600PC (VWR, Radnor, PA, USA) in a quartz cuvette. DLS and zeta-potential measurements were performed with a Zetasizer (Malvern Panalytical, Malvern, UK) in PMMA cuvettes and folded capillary cells. The nanoparticles were dispersed in ethanol for DLS measurements and in water (pH 6.5) for zeta-potential measurements. TEM micrographs were recorded by using a LEO912 (ZEISS, Oberkochen, Germany) microscope operated at an acceleration voltage of 120 kV. A carbon film supported by a standard copper grid was used as sample carrier for TEM characterization. PXRD data were collected in sealed glass capillaries (d = 0.9 mm) with a Stadi-P (Stoe, Darmstadt, Germany) running with Mo-K$_\alpha$ ratiadation and a Mythen detector in Debye–Scherrer geometry.

### 2.1. Synthesis of SiO₂-NH₂

Monodisperse core SiO₂-NP were prepared by a modified Stöber method [31]. The SiO₂-NP possess –OH on their surface which can be used for further surface modification. Thus, the SiO₂-NP were amine-functionalized using (3-aminopropyl)triethoxysilane (APTES). The amination was carried out by dispersing 500 mg of dry SiO₂-NP in 60 mL ethanol under sonication. In total, 3 mL of APTES was added dropwise and stirred overnight at room temperature. The amine-functionalized silica nanoparticles (SiO₂-NH₂) were isolated via centrifugation (8700 rpm, 10 min), washed three times with ethanol and dried at 60 °C for 12 h.

### 2.2. Synthesis of SiO₂@AuNP

Small, around 4 nm, citrate-stabilized AuNP were obtained by reducing chloroauric acid, $(H_3O)AuCl_4 \cdot 2H_2O$, abbreviated as "$HAuCl_4$", with sodium borohydride ($NaBH_4$) in the presence of sodium citrate [32]. Exchanging the citrate ligands with amine groups, the AuNPs were immobilized on the $SiO_2$-NP surface to form $SiO_2$@AuNP seed particles. Therefore, 25 mg of $SiO_2$-$NH_2$ was dispersed in water, 15 mL of the AuNP solution was slowly added and the mixture was allowed to react at room temperature. After centrifugation (8700 rpm, 15 min), an excess of unreacted AuNP was removed and the $SiO_2$@AuNP were washed three times with water. Redispersed in $H_2O$ (5 mg/mL), the particles act as seeds for the core–shell synthesis.

### 2.3. Synthesis of SiO₂@AuNS

The star-shaped growth was obtained by adding various amounts (2.5, 5.0, 75.0, 1000 μL) of a solution of $SiO_2$@AuNP seeds to a solution containing 10 mM PVP (MW = 24,000) in DMF and 82 μL of 50 mM $HAuCl_4$ in aqueous media. The core–shell particles were stirred for 1 h and collected by centrifugation (8700 rpm, 15 min). The $SiO_2$@AuNS were washed three times with ethanol to remove an excess of PVP and redispersed in ethanol (7 mL).

## 3. Results and Discussion

The functionalized nanoparticles were characterized with suitable analytical methods using transmission electron microscopy (TEM), dynamic light scattering (DLS), zeta-potential measurements, UV-Vis measurements and powder X-ray diffraction (PXRD). The presence of the primary amine was confirmed via the use of the Ninhydrin test. TEM was performed to obtain the particle size distribution and revealed that the mean particle size of the prepared $SiO_2$-$NH_2$ nanoparticles to be $81.8 \pm 7.6$ nm. The zeta-potential is a measure of the magnitude of electrical repulsive forces between particles and is predictive of colloidal stability. Particles with zeta-potential greater than +25 mV or less than −25 mV typically have high degree of stability. This is described by the classical Derjaguin, Landau, Verwey, Overbeek (DVLO) theory of colloidal interaction, which predicts that in the absence of van der Waals attraction, the interaction between like-charged colloidal particles is repulsive [33,34]. The value of measured zeta-potential for the $SiO_2$-$NH_2$ nanoparticles is +19.8 mV which indicates a stable colloidal system that is absence of coagulation. In order to investigate the influence of sulfur with higher binding affinity to the AuNP, we used (3-mercaptopropyl)triethoxysilane (MPTS) to functionalize the $SiO_2$-NP and attach AuNP in the second step. No significant change was observed and the typical AuNP coverage of 30% that is reported in literature was not exceeded [35]. The successful synthesis of $SiO_2$@AuNP was verified by TEM and PXRD. Amorphous $SiO_2$-NP show only one broad diffuse peak at a value of 2θ of 10.7°, whereas for $SiO_2$@AuNP, five additional peaks appeared at 2θ of 17.3°, 19.7°, 28.5°, 33.6° and 35.1° which correspond to the reflections of the (111), (200), (220), (311) and (222) crystal planes of Au (see ESI Figure S1). The values are in good agreement with the theoretical pattern calculated from single-crystal data for the face-centered cubic structure of metallic Au with the space group *Fm-3m*.

For the star-shaped nanoshells, a color change from red to purple, blue or green was observed as a function of the concentration of the seeds (see Figure 2).

The resulting $SiO_2$@AuNS showed much stronger plasmonic resonance compared to the $SiO_2$@AuNP or AuNP and were synthesized in high yields (see ESI Figure S2). The nanostructures exhibit high stability over several months. The extinction spectra show slight red-shifts of the maxima associated with narrower bands (see ESI Figure S3).

Coating of the $SiO_2$@AuNP requires a sufficient amount of $HAuCl_4$ which is dependent on the surface area of the core $SiO_2$-NP. By adjusting the $[HAuCl_4]/[seed]$ mass ratio (*R*), we were able to tune $SiO_2$@AuNS growth and the resulting properties. Table 1 summarizes the extinction maximum ($\lambda_{max}$), the diameter ($d_{Au}$ and $d_{hyrd}$) of the growing gold nanostructures and the zeta-potentials ($\zeta$) for different *R* values.

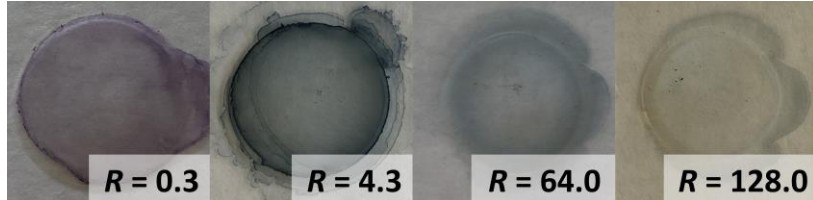

**Figure 2.** Colors of the synthesized gold nanostructures SiO$_2$@AuNS immobilized on a glass substrate depending on the initial seed concentration. The color change indicated a red-shift of the plasmon band with increasing *R*.

**Table 1.** Overview of [HAuCl$_4$]/[seed] ratio (*R*), extinction maxima ($\lambda_{max}$), the size of the gold nanostructure (d$_{Au}$) as well as the hydrodynamic diameter (d$_{hydr}$) and the zeta-potential ($\zeta$) of the SiO$_2$–NH$_2$ nanoparticles or rather SiO$_2$@AuNP before shell formation and coated SiO$_2$@AuNS with varied amounts (1000–2.5 µL) of SiO$_2$@AuNP seed solution.

| | *R* | $\lambda_{max}$ (nm) | d$_{Au}$ (nm) [1] | d$_{hydr}$(nm) [2] | $\zeta$ (mV) |
|---|---|---|---|---|---|
| SiO$_2$-NH$_2$ | - | - | - | - | +19.8 ± 0.1 |
| SiO$_2$@AuNP | - | 515.0 | 3.9 ± 1.2 | - [4] | −18.0 ± 0.9 |
| SiO$_2$@AuNS | 0.3 | 567.0 | 7.1 ± 0.9 | - [4] | −9.1 ± 2.0 |
| | 4.3 | 730.5 | 26.1 ± 6.7 | - [4] | −24.1 ± 1.7 |
| | 64.0 [3] | 814.0 | 107.8 ± 12.8 | 112.4 ± 1.1 | −28.2 ± 1.1 |
| | 128.0 | 879.5 | 146.4 ± 17.2 | 147.3 ± 2.7 | −29.4 ± 0.8 |

[1] Characterized by TEM measurements. [2] Characterized by DLS measurements. [3] Critical ration ($R_{min}$). [4] No spiky shell formed.

The zeta-potentials presented in Table 1 show decreasing surface loadings depending on the stadium of the shell formation. While SiO$_2$-NH$_2$ show a positive zeta-potential caused by free amine groups, negative values were determined for the gold-decorated nanostructures. At higher *R*, i.e., higher gold loadings, lower zeta-potentials were observed. The amount of free amine functionalities decreases when they are linked to the gold surface during the immobilization of gold nanostructures. The more gold is bound to the SiO$_2$-NH$_2$ functions, the less free amine groups are detected, resulting in a negative zeta-potential.

The shell growth can not only be tracked by zeta potential measurements. Figure 3a shows the extinction spectra of the immobilized SiO$_2$@AuNP seeds (dashed line) and the resulting SiO$_2$@AuNS nanostructures (solid lines) depending on different *R*.

Corresponding TEM images of the nanostructures are presented in Figure 3b in order to monitor the actual shape of the formed nanostructure. By increasing *R*, the SPR bands exhibit a red-shift due to the growth of the gold nanostructure. Strong bathochromic shifts indicate the formation of sharp edges, which is well established for AuNS [24]. These growing steps of the gold nanostructures could be observed by TEM micrographs. High seed concentrations (*R* = 0.3, 4.3) only promoted the growth of AuNP but not the formation of a continuous crystalline metallic shell surrounding the SiO$_2$-NP. Above a critical [HAuCl$_4$]/[seed] mass ratio ($R_{min}$) at circa 64.0, a core–shell structure was formed. Further variation in *R* leads to different sized SiO$_2$@AuNS. Therefore, the average nanostructure diameters were determined from the TEM images. At a high SiO$_2$@AuNP seed concentration, equaling low *R*, the AuNP diameter d$_{Au}$ increased from 3.9 nm initial seed diameter to 7.1 nm. The extinction spectra (Figure 3a, red line) show an equally shaped, slightly red-shifted SPR band compared to the initial seeds indicating the growth of gold nanospheres. For lower SiO$_2$@AuNP concentrations, equaling higher *R*, an anisotropic growth occurs. When *R* is increased to a value of 4.3, branched nanostructures of 26.1 nm in diameter (green line) were formed. Extinction spectra and TEM images of three more nanostructures with *R* of 0.6, 3.2 and 12.8 line up perfectly in the trend shown here (see ESI Figures S4-1–S4-3). For *R* of 0.6 and 3.2, AuNP growth up to 20.4 nm is observed, with the subsequent change in the morphology of the nanoparticles (loss of spherical shape). Star-shaped particles were observed at a *R* = 12.8 (d$_{Au}$ = 46.0 nm) with an extinction band

center at 766.0 nm. The nanostructures start to cover the $SiO_2$-NP, but a non-uniform shell was formed. A stronger bathochromic shift is observed due to the SPR resonance of the formed sharp edges. By exceeding $R_{min}$, the star-shaped shells are formed (blue lines) and cause a strong red-shift of the SPR bands up to 879.5 nm indicating the formation of highly branched core–shell structures. Therefore, the SPR of the $SiO_2$@AuNS is found to be in the NIR region, similar to those of gold nanorods (AuNR) and $SiO_2$@Au shells [14,36]. The size of the $SiO_2$@AuNS is tuned by varying the ratio R. For $R = 64.0$, $SiO_2$@AuNS with a diameter of 108 nm were generated with a core diameter of 84 nm. Doubling the ratio produces $SiO_2$@AuNS with a diameter of 146 nm with a core diameter of 112 nm (see ESI Figures S5-1 and S5-2). DLS measurements confirmed the size distribution of the formed $SiO_2$@AuNS (see ESI Figure S6). With 112.4 nm and 147.2 nm, the hydrodynamic diameters are in good agreement with the diameters determined by TEM (see Table 1). Compared to the initial 81.8 nm $SiO_2$ cores, the smaller $SiO_2$@AuNS ($R = 64.0$) are coated with a gold shell of around 13 nm while a shell thickness of 32 nm was obtained for $R = 128.0$ including the tips of the spiky shell. Furthermore, broader extinction bands are observed for the core–shell nanostructures compared to the decorated cores. This observation can be traced back to the interaction of the spikes when a continuous shell is formed [37]. However, the $SiO_2$@AuNS show defined SPR centers in the desired NIR region and a significant trend correlated to the growing of the metallic nanostructures.

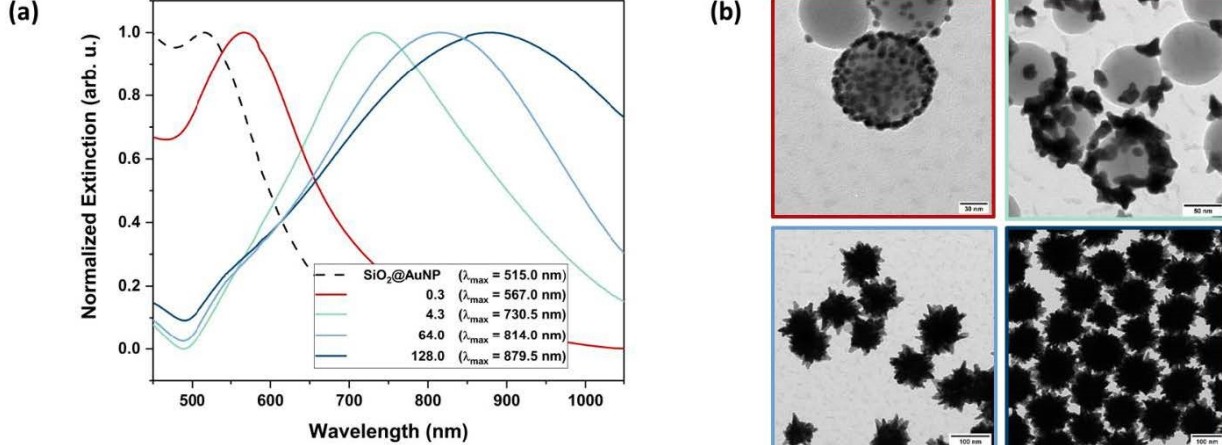

**Figure 3.** Characteristics of the formed nanostructures via UV-Vis and TEM. Extinction spectra (**a**) of the immobilized AuNP seeds (dashed line) and the resulting $SiO_2$@AuNS nanostructures after reaction with different $[HAuCl_4]/[seed]$ mass ratios (solid lines). The according TEM images (**b**) show the stepwise growth of the gold nanostructure on the $SiO_2$-NP surface and are color-coded with respect to the UV-Vis spectra. The shell formation is observed for R lager than 64.

## 4. Conclusions

In conclusion, we have developed a reliable synthesis of easily accessible gold nanoshell formation on $SiO_2$ cores using the PVP/DMF reduction method. We succeeded in synthesizing $SiO_2$@AuNS core–shell nanoparticles of high stability with tunable shell thicknesses. The synthesis proceeded under mild conditions with short reaction times and simplifies the gold coating procedure significantly. This approach combines the advantages of $SiO_2$@Au and AuNS systems resulting in a perfect overlap of the SPR bands of the star-shaped gold nanostructures with the first biological window, providing a promising platform for plasmonic-enhanced NIR excitation for biological applications, e.g., in combination with upconversion nanoparticles. Beyond that, the coating approach is accessible for all $SiO_2$-coated nanostructures and therefore it is highly versatile as most of the nanomaterials can be coated with a $SiO_2$ shell.

**Supplementary Materials:** The following are available online at https://www.mdpi.com/article/10.3390/chemistry4030046/s1. Figure. S1 PXRD pattern of $SiO_2$(82nm) in black, $SiO_2$(82nm)@AuNP in red and the theoretical pattern for the face-centred cubic structure of metallic Au with the space group *Fm-3m*. Figure. S2 TEM image sections of $SiO_2$@AuNS($R$ = 128.0) demonstrating the high ratio of coated $SiO_2$ nanoparticles with 4,000 times magnification (a) and 10,000 times magnification (b). Figure. S3 TEM images of $SiO_2$@AuNS($R$ = 64.0, light blue) (a), and $SiO_2$@AuNS($R$ = 128.0, dark blue) (b) stored f*or* seven month in ethanol and corresponding extinction spectra of the nanostructures (c). Solid lines show the initial spectra while dashed lines mark the spectra of the nanostructures recorded aft*er* seven month. Figure. S4–1 Extinction spectrum and TEM image of the $SiO_2$@AuNS fo*r* $R$ = 0.6 (500.0 µL of the seed solution) with a gold nanostructure size of 11.3 ± 1.9 nm. Figure. S4–2 Extinction spectrum and TEM image of the $SiO_2$@AuNS fo*r* $R$ = 3.2 (100.0 µL of the seed solution) with a gold nanostructure size of 20.4 ± 4.6 nm. Extinction spectrum and TEM image of the $SiO_2$@AuNS for $R$ = 12.8 (25.0 µL of the seed solution) with a gold nanostructure size of 46.0 ± 12.8 nm. Figure. S4–3 Extinction spectrum and TEM image of the $SiO_2$@AuNS for $R$ = 12.8 (25.0 µL of the seed solution) with a gold nanostructure size of 46.0 ± 12.8 nm. Figure. S5–1 Histograms of the core and the total diamete*r* of $SiO_2$@AuNS (C) with a mass ratio of 64.0 (5.0 µL of the seed solution) determined by TEM. Figure. S5–2 Histograms of the core and the total diamete*r* of $SiO_2$@AuNS (D) with a mass ratio of 128.0 (2.5 µL of the seed solution determined by TEM). Figure. S6 Normalized number-averaged DLS profiles of $SiO_2$@AuNS ($R$ = 64.0) and $SiO_2$@AuNS ($R$ = 128.0) dispersed in ethanol.

**Author Contributions:** L.C.S. synthesized and characterized the compounds and wrote the first draft; M.S.W. and J.A.C. reviewed and supervised the work. All authors have read and agreed to the published version of the manuscript.

**Funding:** This research received no external funding.

**Institutional Review Board Statement:** Not applicable.

**Informed Consent Statement:** Not applicable.

**Data Availability Statement:** Additional data can be found in the supporting informations. Primary data are available from the authors upon request.

**Acknowledgments:** L.C.S. gratefully acknowledges the support from the German Academic Exchange Service (DAAD) and MITACS. J.A.C. is a Concordia University Research Chair in Nanoscience and is grateful to Concordia University for financial support. J.A.C. is grateful to the Natural Science and Engineering Research Council (NSERC) of Canada for the sustained support of his research. We thank Silke Kremer for recording the powder diffraction data and Tabea von Keitz for technical assistance.

**Conflicts of Interest:** The authors declare no conflict of interest.

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
