# Peer review of "Growing Gold Nanostars on SiO2 Nanoparticles: Easily Accessible, NIR Active Core–Shell Nanostructures from PVP/DMF Reduction"

_chemistry, doi:10.3390/chemistry4030046_

Round 1

Reviewer 1 Report

The authors have presented a new synthetic strategy towards a controlled gold/silica core-shell nanostructure system. The authors have presented an acceptable amount of data collected with clear discussions on the findings supporting the conclusions arrived in this manuscript. The approach and the manuscript content well align with the journal scope. Thus, the reviewer recommends the manuscript.

Reviewer 2 Report

Dear Authors,

the paper is interesting and most of the results appeared to be promising there a few notes that could enhance the quality in somehow:

the material characterization should be mentioned in the abstract such as TEM, DLS, etc.

keywords should be arranged alphabetically.

fig 1 can be moved to the section of materials and methods.

more information regarding the PXRD analysis.

the author may add a comparison between the obtained  results and other research in  gold nanorod (or any other morphology) on SiO2 nanoparticles.

Regards 

Reviewer 3 Report

Detailed comments are as follows:
1.The title is not catchy and does not reflect essential contents.
2.Where is the practical application of this manuscript? It must be added.
3. Check the grammar throughout the article and correct it. Proofread the article as many language errors were identified.
4. The introduction needs to be further revised to highlight the purpose of the study, You need to introduce what others have studied and what needs further research. Besides, the following all of references are recommended to be cited:
Green synthesis of DyBa2Fe3O7. 988/DyFeO3 nanocomposites using almond extract with dual eco-friendly applications: Photocatalytic and antibacterial activities// Synthesis, characterization and application of Co/Co3O4 nanocomposites as an effective photocatalyst for discoloration of organic dye contaminants in wastewater and antibacterial properties///Photo-degradation of organic dyes: simple chemical synthesis of Ni(OH)2 nanoparticles, Ni/Ni(OH)2 and Ni/NiO magnetic nanocomposites
5. The conclusion must be more than just a summary of the manuscript.
6. The figure captions should be written with more informative.
7. What is the main purpose of TEM images?
8. Why the author doesn't measure the size distribution and zeta potential using DLS? If there is a possibility please measure them.
9. The quality of figure 2 is not acceptable.
10. The relevance/novelty of the work needs to be highlighted.

Round 2

Reviewer 3 Report

Accept